# Viruses in Simuliidae: An Updated Systematic Review of Arboviral Diversity and Vector Potential

**DOI:** 10.3390/life15050807

**Published:** 2025-05-19

**Authors:** Alejandra Rivera-Martínez, S. Viridiana Laredo-Tiscareño, Jaime R. Adame-Gallegos, Erick de Jesús de Luna-Santillana, Carlos A. Rodríguez-Alarcón, Julián E. García-Rejón, Mauricio Casas-Martínez, Javier A. Garza-Hernández

**Affiliations:** 1Instituto de Ciencias Biomédicas, Universidad Autónoma de Ciudad Juárez, Juárez 32310, Chihuahua, Mexico; alejandra.riveramtz0@gmail.com (A.R.-M.); viridiana.laredo@gmail.com (S.V.L.-T.); carrodri@uacj.mx (C.A.R.-A.); 2Facultad de Ciencias Químicas, Universidad Autónoma de Chihuahua, Chihuahua 31125, Chihuahua, Mexico; jadame@uach.mx; 3Laboratorio Medicina de la Conservación, Centro de Biotecnología Genómica del Instituto Politécnico Nacional, Reynosa 88710, Tamaulipas, Mexico; 4Laboratorio de Arbovirología, Centro de Investigaciones Regionales “Dr. Hideyo Noguchi”, Universidad de Yucatán, Mérida 97225, Yucatán, Mexico; julian.garcia@correo.uady.mx; 5Centro Regional de Investigación en Salud Pública, Instituto Nacional de Salud Pública, Tapachula 30700, Chiapas, Mexico; mcasas@insp.mx

**Keywords:** black flies, Simuliidae, virus, arbovirus, metagenomics

## Abstract

Black flies (Diptera: Simuliidae) are important vectors of pathogens, including filarial nematodes, protozoans, and arboviruses, which significantly impact human and animal health. Although their role in arbovirus transmission has not been as thoroughly studied as that of mosquitoes and ticks, advances in molecular tools, particularly metagenomics, have enabled the identification of non-cultivable viruses, significantly enhancing our understanding of black-fly-borne viral diversity and their public and veterinary health implications. However, these methods can also detect insect-specific viruses (i.e., viruses that are unable to replicate in vertebrate hosts), which may lead to the incorrect classification of black flies as potential vectors. This underscores the need for further research into their ecological and epidemiological roles. This systematic review, conducted following the PRISMA protocol, compiled and analyzed evidence on arbovirus detection in Simuliidae from scientific databases. Several arboviruses were identified in these insects, including vesicular stomatitis virus New Jersey serotype (VSVNJ), Venezuelan equine encephalitis virus (VEEV), and Rift Valley fever virus. Additionally, in vitro studies evaluating the vector competence of Simuliidae for arboviruses such as dengue virus, Murray Valley encephalitis virus, and Sindbis virus were reviewed. These findings provide critical insights into the potential role of black flies in arbovirus transmission cycles, emphasizing their importance as vectors in both public and veterinary health contexts.

## 1. Introduction

Black flies (Diptera: Simuliidae) are significant vectors of public health importance due to their ability to transmit pathogens affecting both humans and animals, such as onchocerciasis and mansonellosis; however, their potential as vectors of medically important viruses remains largely understudied [1]. Improving our knowledge of their capacity to transmit arboviruses could have profound implications for public health, particularly in regions where these vectors are abundant and where associated diseases may be underreported or misdiagnosed [2]. This is attributable to several factors, including the technical challenges associated with collecting these insects from their often restricted and remote habitats, the difficulty of establishing and maintaining stable colonies under laboratory conditions, and the limited feasibility of conducting vector competence studies [3,4] (this issue is discussed in more detail in a subsequent section).

Traditionally, the detection of arboviruses in vectors has relied on virus isolation through cell culture and nucleic acid detection using molecular biology techniques [5]. While these methods are valuable, they come with limitations, particularly in identifying non-cultivable or low-prevalence viruses, which leads to an underestimation of viral diversity in blood-feeding vectors [6]. In recent years, the advent of next-generation sequencing (NGS) and metagenomics has provided a powerful alternative for more precise and comprehensive virus detection without the need for in vitro cultures [7].

Metagenomics has revolutionized pathogen research by enabling the identification of previously undetectable microorganisms, facilitating rapid genome sequencing, and uncovering pathogens of both human and veterinary importance. This technology has improved scientific expansion in fields such as taxonomy, microbial ecology, evolutionary biology, and, most notably, epidemiological surveillance and the discovery of novel zoonotic pathogens [8,9]. By allowing for the high-throughput, unbiased screening of biological samples, Metagenomics can reveal the full scope of viral diversity present within vector populations, providing insights into viral ecology and transmission dynamics [10]. Despite the rapid progress that has been made in applying metagenomic tools to study viral diversity in other hematophagous insects, such as mosquitoes and biting midges, the research focused on Simuliidae remains limited. This gap in knowledge underscores the need for further investigation, as black flies may harbor viruses of epidemiological relevance that have gone undetected due to methodological constraints in previous studies.

Therefore, the present systematic review aims to compile and critically analyze the available evidence on arbovirus detection in Simuliidae using molecular biology and metagenomic approaches. By providing a comprehensive overview of the current state of knowledge, this study seeks to enhance the understanding of viral diversity in these hematophagous dipterans and their potential role in the transmission of arboviruses relevant to humans, animals, and wildlife.

## 2. Materials and Methods

To conduct this systematic review, the PRISMA (Preferred Reporting Items for Systematic Reviews and Meta-Analyses.) protocol was followed, to ensure compliance with each step to identify, analyze, select, and incorporate the relevant literature [11] (Figure 1). The search was performed using academic search engines and databases such as PubMed, Google Scholar, Google Books, and ScienceDirect. Both English and Spanish terms were used, with a carefully selected set of keywords including “Simuliidae”, “*Simulium*”, “virome”, “virus”, “metagenomics”, “arbovirus”, “black flies”, “horizontal transmission”, “biological transmission”, “mechanical transmission”, “vector competence”, “vector competence barriers”, and “vector”. No restrictions were applied regarding publication years, allowing the inclusion of literature from the 20th century up to 5 May 2025.

## 3. Results Overview

### 3.1. Simuliidae as Vectors of Public-Health-Related Non-Viral Pathogens

The Simuliidae family, or black flies, includes approximately 2163 morphospecies of hematophagous dipterans globally [12]. Only females require blood meals, making them the primary vectors for pathogen transmission [13]. Host preferences vary, with species exhibiting zoophilic, anthropophilic, or opportunistic feeding behaviors. This behavior has significant medical and socioeconomic impacts, as high biting rates reduce tourism, cause animal mortality, and facilitate the spread of pathogens [3]. Notable pathogens transmitted by Simuliidae include *Onchocerca volvulus*, the causative agent of human onchocerciasis [14,15], and *Mansonella ozzardii* [1,16]. In veterinary medicine, Simuliidae are key vectors of *Onchocerca* spp., causing filariasis in livestock, dogs, wild ungulates [17], and equids [18]. They also transmit *Dirofilaria ursi* in black bears (*Ursus americanus* and *Ursus thibetanus japonicus*), with occasional human infections [19,20], and *Splendidofilaria fallisensis* in ducks and waterfowls [21]. Additionally, they spread protozoa such as *Trypanosoma* spp. and *Leucocytozoon* spp. in domestic and wild birds [22].

Simuliidae also contribute to public health challenges through severe blackfly hypersensitivity (SBH), a condition resulting from repeated bites in humans and animals [23,24]. SBH manifests as skin inflammation, intense itching, and prolonged rashes (Figure 2). Immune responses to salivary toxins can lead to systemic symptoms, including fever, headaches, and fatigue [25].

### 3.2. General Relevance of Arbovirosis in Vector-Borne Diseases

The term “arbovirus”, derived from arthropod-borne viruses, refers not to a taxonomic classification but to a group of viruses defined by their transmission mechanism [26]. These viruses rely on infected vertebrate hosts, who are bitten by hematophagous arthropods such as mosquitoes, ticks, *Culicoides*, and black flies, etc. Vectors acquire the virus during feeding, replicate it, and transmit it to new hosts, a process known as biological horizontal transmission [27,28]. Additional biological transmission routes include vertical (transovarial) transmission, in which larvae emerge from the eggs of infected females, and venereal transmission, where viremic females transmit the virus to males during mating [29,30].

Arboviral diseases have historically been recognized as a global public health concern, with a notable rise in cases during the late 20th and early 21st centuries due to their emergence and re-emergence [31]. Their spread is driven by climatic, ecological, and sociodemographic factors [32,33], as well as human activities that cause ecological disruptions [34]. These factors increase disease prevalence, particularly in socially disadvantaged regions with limited healthcare access and inadequate pathogen monitoring [35]. Urban expansion, population growth, and poor urban planning further exacerbate the spread. Migration and informal settlements heighten susceptibility to arboviruses, straining infrastructure, healthcare, and food systems [36]. These ecological and demographic shifts have a significant impact on arbovirus epidemiology.

In recent decades, the increasing prevalence and global spread of arboviruses have come to represent a significant public health threat, causing high morbidity and mortality in humans and animals [26]. Over 500 arboviruses have been documented (https://wwwn.cdc.gov/arbocat/ accessed on 5 July 2024); these are classified into nine families (*Flaviviridae*, *Togaviridae*, *Peribunyaviridae*, *Phenuiviridae*, *Nairoviridae*, *Sedoreoviridae*, *Orthomyxoviridae*, *Rhabdoviridae*, and *Asfarviridae*). Table 1 summarizes the key molecular characteristics of a representative sample of arbovirus families.

### 3.3. Detection and Isolation of Viruses in Simuliidae

Studies on the detection and isolation of arboviruses in Simuliidae remain limited and scarce. Nevertheless, the existing research has demonstrated that certain species within this family can act as vectors for arboviruses of significant public and veterinary health importance. Black flies have been implicated in the transmission of several arboviruses, including Leporipoxvirus (family *Poxviridae*), which primarily infects rabbits and squirrels [53,54], as well as various serotypes of vesicular stomatitis virus (VSV).

VSV, belonging to the genus Vesiculovirus within the family *Rhabdoviridae*, has a global distribution and is endemic to regions such as North America, Central America, northern South America, and parts of the southeastern United States, primarily affecting livestock, including horses, mules, donkeys, cattle, and pigs [55,56]. Consequently, it is considered the most important animal health arbovirus transmitted by Simuliidae. Notably, cases of VSV have been detected in livestock in regions of Mexico with high black fly prevalence [57,58]. These findings underscore the role of Simuliidae in the epidemiology of VSV and highlight the need for further research into their vector capacity for other arboviruses; the most recent findings in relation to this virus are discussed later in the text.

Several experimental studies have contributed to understanding the transmission dynamics of arboviruses by black flies. In 1934, Knowlton attempted to demonstrate the transmission of equine encephalitis virus from a horse to a guinea pig; however, the experiment did not yield conclusive evidence of successful transmission [59]. In contrast, Ferris et al. (1955) [60] demonstrated the ability of black flies to mechanically transmit VSV, although only for a short period following feeding on infected embryos. Transmission was effective only within the first 24 h, with no evidence of viral replication or an incubation period within the flies. As a result, their role as vectors was attributed to contamination of the mouthparts, classifying them as potential mechanical vectors, and not as biological vectors.

Historically, Cupp et al. (1992) [57] provided the first confirmed evidence of the biological transmission of an arbovirus by a member of the Simuliidae family. In their study, they evaluated the competence of *Simulium vittatum Zetterstedt* to act as a vector for the New Jersey serotype of vesicular stomatitis virus (VSVNJ). To achieve this, they infected female flies of this species via intrathoracic inoculation and oral exposure using a strain obtained during the 1982 VSVNJ outbreak in North America. Among the reported results, 70% of thoracically infected flies and 45% of orally infected flies secreted infectious virions in their saliva 10 days post-infection, indicating that VSVNJ exhibits positive vector competence. Further significant findings regarding the biological transmission of arboviruses occurred in 1953, when the Rift Valley fever virus was isolated from *Simulium* sp. collected near the Orange River in South Africa. This discovery suggested that, in addition to mosquitoes, other arthropods could play a role in the transmission of this arbovirus in nature [61,62].

Previously, Austin (1967) [63] evaluated the susceptibility for the biological and mechanical transmission of ten arboviruses in the blackfly *Austrosimulium ungulatum* through intrathoracic inoculation and feeding on viremic mice. The viruses examined included Semliki Forest virus (SFV), Whataroa (M78), Sindbis (MRM39), Bebaru (AMM2354), Murray Valley encephalitis (MVE), St. Louis encephalitis (SLE), Ntaya, Dengue I, Dengue II, and Batai. At serial post-infection intervals maintained at 22 °C, viral replication was assessed in the heads, salivary glands, and bodies of the black flies. The successful replication of the M78 virus was detected exclusively in the body from six days post-infection, with significantly higher viral titers compared to the baseline levels observed in control flies sacrificed immediately after infection. However, M78 was found to be refractory, indicating that viral dissemination or replication into the salivary glands did not occur, consequently, the virus was not transmitted. Similarly, the SFV, MRM39, AMM2354, Ntaya, and Chittagong viruses persisted for at least 10 days post-infection, whereas Dengue I, Dengue II, and SLE viruses were cleared before day 14 and were not detected in the heads (salivary glands). Therefore, it was determined that there is no possibility of the arboviruses tested being biologically transmitted.

However, the mechanical transmission of these arboviruses was also evaluated by feeding groups of *A. ungulatum* first on viremic suckling mice, and then on no viremic suckling mice. They found that *A. ungulatum* was able to transmit the M78 virus for up to 48 h via mechanical transmission under experimental conditions following the ingestion of viremic blood. These results support the hypothesis that, under specific conditions such as a high viral load and close contact with infected vertebrate hosts, black flies may contribute to the mechanical transmission of certain arboviruses, even if they are not competent biological vectors. Later, Sanmartín et al. (1973) [64] investigated the role of black flies in the transmission of Venezuelan equine encephalitis virus (VEEV) during outbreaks recorded in 1967 in three localities in Colombia. In two locations, high black fly activity was identified, particularly among *Simulium exiguum*, *S. metallicum*, *S. callidum*, *S. paynei*, and *S. mexicanum*. The virus was successfully isolated in eleven samples from those specimens collected during the biting of horses and humans by intracerebrally inoculating two-to-seven-day-old white mice or by inoculating Vero cells. Except for the *S. paynei* pools, all specimens were macerated immediately after collection. This introduces potential bias in the interpretation of results, leading to the erroneous incrimination of black flies as the true biological vectors, because the viral isolation involved recently ingested blood from viremic hosts rather than from an active infection within the black flies. In contrast, due to the low post-collection mortality of *S. paynei*, these specimens were not processed immediately but were kept alive for several days prior to analysis, suggesting possible virus replication within the insect. In subsequent studies, the vector competence of *S. mexicanum* and *S. metallicum* for enzootic and epizootic Colombian strains of VEEV was experimentally assessed [65]. Wild-caught flies fed on viremic guinea pigs were monitored for viral replication and tested for biological and mechanical transmission. No evidence of biological transmission or sustained viral replication was found, though occasional mechanical transmission occurred. These findings suggest that neither species plays a significant role in VEEV transmission in nature. Notably, Gibbs and Long (2013) [66] report that VEEV was isolated from various families of hematophagous Diptera, including Simuliidae, referencing a study by Mitchell et al. (1985) [67]. However, upon reviewing the original study by Mitchell et al. (1985) [67], which involved prospective arbovirus surveillance in three provinces of Argentina between 1977 and 1980, a total of 104 Simuliidae specimens were tested for viruses, and all pools were negative. Therefore, the conclusion by Long and Gibbs (2013) [66] appears to be incorrect.

A discovery by Sommerman (1977) [68] demonstrated the first evidence of snowshoe hare virus (SSH) virus in *Simulium* sp., suggesting a broader vector range and highlighting the need for further investigation into the role of black flies in arbovirus transmission. In this study, the SSH virus, a member of the California serogroup, was isolated from multiple mosquito species and one unidentified black fly collected in interior Alaska between 1970 and 1972. Then, during an arbovirus survey conducted in Colorado and Utah amid the 1982–1983 epizootic of vesicular stomatitis New Jersey virus (VSNJ) in the western United States, a bunyavirus sequence identified as Lorken virus (Bunyamwera group) was detected in a pool of *Simulium bivittatum* collected near a horse livestock area and adjacent mountain sagebrush and alfalfa pasture [69].

Chanteau et al. (1993) [70] conducted a study on Nuku-Hiva Island, in the Marquesas Archipelago of French Polynesia, to investigate the potential role of *Simulium buissoni* in the epidemiology of hepatitis B virus (HBV) within a holoendemic population. Although there was no evidence of viral replication within the black flies, HBV DNA was detected only in small amounts in a few black fly pools, with a low viral load. Additionally, a positive correlation was observed between the number of skin lesions caused by black fly bites and HBV infection prevalence—particularly among children living in rural areas. This supports the hypothesis that, in regions with high black fly density, HBV transmission might occur indirectly through bite-induced skin lesions, without requiring viral replication within the insects.

Multiple studies have investigated the role of black flies in the transmission of vesicular stomatitis viruses. Mead et al. (1997) [71] evaluated the vector competence of several black fly species in southern Arizona for different isolates of vesicular stomatitis VSNJV. Their findings demonstrated that *S. vittatum* served as a competent vector, while other species, including *S. bivittatum* and *Simulium longithallum*, were refractory to infection. Expanding on these finding, Mead et al. (1999) [72] provided experimental confirmation of VSNJV transmission by *S. vittatum* using a murine model. In this study, artificially infected colony-reared flies were allowed to feed on susceptible mice, all of which developed neutralizing antibodies by 21 days post-infection, confirming successful transmission.

Subsequently, Mead et al. (2000) [73] expanded their research to assess the competence of both colonized and wild-caught black fly species for the VSV-Indiana serotype. They demonstrated that both *S. vittatum* and *Simulium notatum* were competent vectors, as an infectious virus was detected in the saliva of exposed flies, suggesting their potential role in the transmission of this serotype. Further validation of biological transmission was achieved by Mead et al. (2004) [74], who confirmed VSNJV infection in domestic pigs following exposure to infected black flies, supporting the vectorial capacity of VSN in Simuliidae in natural host systems. Supplementary field evidence was later provided by Drolet et al. (2006) [75] during a VSNJV outbreak in Wyoming, where large numbers of black flies were collected and screened for viral RNA. *S. bivittatum* was identified as a potential vector, with viral RNA detected in two pools, suggesting its involvement in outbreak dynamics under natural conditions. Mead et al. (2009) [76] reported the first successful transmission of VSNJV to cattle by infected black flies, resulting in clinical disease. Infected flies fed at sites where VSNJV lesions are typically observed, such as the mouth and coronet band, leading to local viral replication and vesicular lesions in cattle. Similarly, Howerth et al. (2002) [77] conducted experiments to determine how black flies become infected with VSNJV. Their findings indicated that the virus initially infects the gut, but transmission to the salivary glands may be blocked in older flies, impairing their ability to transmit the virus. This study also highlighted the role of black fly age in determining vector competence.

Recently, Mesquita et al. (2017) [78] explored the role of rodents, specifically, juvenile and nestling deer mice (*Peromyscus maniculatus*), in VSNJV transmission. They exposed these mice to VSNJV-infected *S. vittatum* bites and observed the development of severe neurological symptoms in some juvenile mice within six to eight days post-inoculation. Interestingly, juvenile mice did not develop viremia, while nestling mice did, and they exhibited widespread viral antigen distribution in their central nervous systems. Furthermore, these viremic nestling mice could infect naive black flies, demonstrating that these rodents could act as potential reservoirs or amplifying hosts for VSNJV.

In both the U.S. and Mexico, recent studies on arbovirus transmission dynamics have assessed the presence of VSV in various species of hematophagous dipterans, including Simuliidae species. Through a molecular analysis, VSVNJ RNA was detected in black fly samples, confirming their role as vectors in the transmission of this arbovirus in the region [55,79]. Similarly, in 2023, Scroggs et al. [80] conducted a large-scale vector surveillance study during a VSNJV outbreak across California, Nevada, and Texas, US. Several specimens of dipterans were collected, including Culicoides and *Simulium* spp., detecting a 96% rate of RNA-positive pools in both species, representing the first report of VSNJV in several wild-caught *Simulium* spp. The study highlights the importance of vector surveillance in understanding VSNJV transmission dynamics and informs control efforts. McGregor et al. (2021) [81] contributed further evidence of black flies and biting midges as VSNJV vectors during a VSV outbreak in the United States. Their surveillance in Kansas identified Culicoides and *Simulium* spp. as important vectors, with positive pools detected for the VSV-Indiana serotype. This study also reported new vector species, expanding the list of potential VSNJV vectors and emphasizing the need for targeted surveillance.

Despite increased efforts regarding arbovirus surveillance in Simuliidae, few studies have documented the presence of other arboviruses associated with this group as potential vectors. For example, Reeves and Milby (1990) [82] examined thousands of *Simulium* spp. in California, USA, but they were unable to detect or isolate any arboviruses, supporting the notion that these flies may possess limited or no vector competence under certain ecological conditions. More recently, Cavallaro et al. (2018) [83] evaluated the effectiveness of partially submerged adhesive traps for the surveillance of adult black flies and the detection of arboviruses. The collected species were identified as *S. vittatum* and *S. decorum*. Although positive cases of West Nile virus were detected in *Culex* spp. mosquitoes collected within a 16 km radius of the study sites, tests conducted on black flies did not reveal the presence of West Nile virus or Eastern equine encephalitis virus. Additionally, during a severe outbreak of bluetongue disease in a flock of sheep in Bruneau, Owyhee County, Idaho, various hematophagous dipterans were collected, including 89 black flies that had been in direct contact with infected animals. However, the virus could not be isolated from any of the collected black flies [84].

In the past decade, researchers have hypothesized that endemic Simuliidae species in African onchocerciasis-endemic regions may play a critical role in transmitting an unidentified virus, referred as Nodding Syndrome Virus (NSV), potentially associated with *O. volvulus*. This virus is suspected to contribute to a newly emerging neurotropic syndrome affecting both humans and other animal species. The syndrome has garnered significant attention within the scientific community due to its debilitating effects and its potential link to onchocerciasis. Such an association could complicate disease control efforts and pose a substantial public health risk in affected regions [85]. However, the exact etiology of onchocerciasis-associated epilepsy remains a subject of ongoing debate [86].

The CDC Arbovirus Catalog (https://wwwn.cdc.gov/arbocat/ accessed on 20 September 2024) [87] records the isolation of the arboviruses mentioned previously, as well as the Jerry Slough virus (JSV), which is transmitted by *Simulium* sp. However, no further information regarding this virus is available. Table 2 details viruses of medical and veterinary significance associated with black flies, along with the methodologies employed in the studies.

### 3.4. Metagenomics of Viromes as a Tool for the Detection of Zoonotic Arboviruses

Over two-thirds of human pathogens are zoonotic in origin [88], with RNA viruses, including arboviruses, constituting the majority of them [89,90]. Historically, unconventional methods for detecting these viruses involved using sentinel animals exposed to arthropod bites in the wild, as exemplified by the discovery of Zika virus in Africa [91]. Post-infection, serological diagnostics or viral isolation in cell cultures were employed [92]. Advances in molecular biology have shifted the focus to viral genome detection using PCR and Sanger sequencing, enabling the identification of non-cultivable arboviruses [93]. However, these methods require prior knowledge of the viral genome, such as oligonucleotide design for targeted amplification.

Next-generation sequencing (NGS), or metagenomics, enables the high-throughput sequencing of all viral genomes in a sample (Figure 3), facilitating the discovery of novel viral species [94]. Since 2010, 86 studies on arbovirus identification in mosquitoes and ticks have been documented on PubMed, with a notable increase in mosquito virome research between 2016 and 2022. These studies identified over 2000 new viruses, including 112 in Chinese arthropods [95], 1445 RNA viruses in multiple Chinese provinces [96], 32 in *Culex* mosquitoes in California [97], and 101 in *Culex* and *Aedes* species in Yunnan, China [98]. A recent study in São Paulo, Brazil, identified 229 viral species in *Aedes*, *Anopheles*, and *Culex* mosquitoes, primarily from the orders *Picornavirales*, *Nodamuvirales*, and *Sobelivirales* [99]. In Mexico, metagenomic studies in Yucatán have revealed RNA and mosquito-specific viruses, including Uxmal virus (*Aedes taeniorhynchus*) and Mayapán virus (*Psorophora ferox*) [100]. Subsequent research identified Houston virus in *Culex quinquefasciatus* [101], Ek Balam virus (*Tymoviridae*) in *C. quinquefasciatus* [102], and novel rhabdoviruses in Culex mosquitoes in the U.S. [103].

These studies of RNA virus isolation in mosquitoes worldwide expand the diversity of viruses discovered through metagenomics. Metagenomic approaches applied to mosquitoes have become key tools for virological surveillance, allowing for the identification of both known and unknown viruses related to diseases transmitted by hematophagous dipterans [104]. However, despite these advances, only two studies of species of the Simuliidae family have been published, as detailed below.

### 3.5. Metagenomic Studies of Viruses in Simuliidae

In the available scientific literature, studies of the metagenomics of arboviruses in Simuliidae are limited. The first such study was conducted by Kraberger et al. (2019) [105] in New Zealand, where 40 *Simulium* specimens from a single site were analyzed. In this study, new viruses belonging to four viral families were identified: *Genomoviridae* (n = 9), *Circoviridae* (n = 1), *Microviridae* (n = 108), and other unclassified viruses (n = 20). However, no in vitro experiments were performed to determine whether any of these viruses have the capacity to infect and replicate in vertebrate cells, leaving their potential as arboviruses unknown. Subsequently, Kobayashi et al. (2020) [106] conducted a pilot study in Japan on the virome of certain Culicomorpha, analyzing only three specimens of *Simulium aureohirtum*. Despite the small sample size, three new viruses were detected, belonging to three viral taxa: *Dicistroviridae*, *Nodaviridae*, and an unclassified taxon. As in the previous study, no vector competence assays were conducted in vertebrate cells. In contrast, De Coninck et al. (2024) [107] analyzed the virome of black flies in Cameroon and detected 641 RNA viruses, identified based on the sequence of RNA-dependent RNA polymerase (RdRP), many of which corresponded to previously unknown viral lineages.

These studies demonstrate that metagenomics is a powerful tool for discovering new viruses and characterizing viromes in insects such as Simuliidae. The integration of this NGS methodology with molecular studies, viral isolation, and vector competence assays in vertebrate cells will allow for a better understanding of the relevant interactions, which will make significant contributions to the advancement of scientific knowledge.

### 3.6. Exploring Causal and Casual Modes of Arbovirus Transmission Mechanisms in Simuliidae

Understanding the mechanisms of horizontal transmission (Figure 4) is fundamental to comprehending the epidemiological dynamics of the causal relationships within vector–pathogen–host interactions. Turell (1988) [108] defined this as the transfer of any virus from an infected arthropod vector to a vertebrate host, typically occurring during blood feeding.

In Simuliidae, the horizontal transmission of arboviruses occurs through two principal modes. Biological transmission (Figure 4A) involves the systemic infection of the vector, with viral dissemination and replication across vector competence barriers, ultimately leading to amplification in the salivary glands for transmission during subsequent blood feeding [109,110]. This process also involves several physiological mechanisms of the vector, including cellular immunity, microbial interactions, enzymes, iRNA, and the genetic background [111]. The alternative mode is mechanical (oral) transmission (Figure 4B), which entails the passive transfer of viral particles from one infected vertebrate host to another via contaminated mouthparts, without viral replication within the vector [112].

As described above, mechanical transmission has been experimentally demonstrated in vitro in *A. ungulatum* and *S. venustum* [60,63], both of which successfully transmitted vesicular stomatitis virus (VSV) to susceptible hosts 24–48 h after ingesting viremic blood. Although feasible, mechanical transmission is highly dependent on elevated viral title levels in the vertebrate host and is generally regarded as incidental or as resulting from casual interactions, with limited significance in natural virus maintenance [113]. However, under epidemic conditions, it may temporarily contribute to the amplification of viral spread [112]. In contrast, biological transmission entails a causal relationship between the vector and the arbovirus. For example, *S. vittatum* has been confirmed as a competent biological vector of VSV [55], with up to 45% of orally infected individuals harboring an infectious virus in their saliva following a 10-day extrinsic incubation period. This capacity for viral replication and subsequent transmission was further validated using murine and swine models, providing robust evidence of true vector competence [72].

Particularly relevant examples for examining whether arbovirus transmission is causal or incidental include RVFV and NSV. In the case of RVFV, *Aedes* mosquitoes (Culicidae) are well-established as the primary vectors; RVFV has been isolated from field-collected *Simulium* specimens in an endemic region of South Africa [61]. This observation has increased speculation about the potential role of black flies in the transmission dynamics of the RVFV. It suggests a possible, though unconfirmed, role of Simuliidae in RVFV transmission cycles, particularly during epizootic occurrences where multiple transmission routes may emerge under favorable environmental conditions [114]. Nonetheless, to date, no study has demonstrated the biological competence of Simuliidae for RVFV, and this association should therefore be interpreted with caution, as it may reflect incidental rather than causal involvement in transmission [115].

Another notable example comes from the emerging evidence in Central Africa, which suggests the possible circulation of a neurotropic virus, commonly referred to as NSV, associated with high black fly biting rates and onchocerciasis-related nodules in endemic regions [85,86,116]. Although a specific etiological agent has not been conclusively identified, one hypothesis proposes that *O. volvulus*, transmitted by *Simulium* spp., may harbor an endosymbiotic virus that is responsible for the observed neurological syndromes. In this scenario, black flies could serve as vectors for both the filarial parasite and the neurotropic virus, thereby contributing to the pathogenesis of NSV [117].

The evidence from both RVFV and NSV cases underscores the urgent need for integrated studies to better understand potential interactions between Simuliidae and arboviruses. In these cases, it is essential to distinguish between the mere detection of viral material and actual vector-mediated transmission. Molecular techniques such as RT-PCR or next-generation sequencing (NGS) can detect viral genetic material without proving that the virus can replicate or be transmitted to new hosts [118]. To establish Simuliidae as true biological vectors, laboratory experiments are needed to demonstrate viral replication, dissemination to the salivary glands, and successful transmission during blood feeding [119].

Clarifying the role of black flies in arbovirus transmission is important for both public health and scientific understanding. An inaccurate assessment or neglect of their potential role as vectors can compromise arbovirus surveillance efforts and limit the effectiveness of vector control programs. Therefore, the limited attention given to Simuliidae in arbovirology reflects a significant gap in our knowledge. Active viral monitoring in black fly populations, especially in areas where outbreaks occur without clearly identified vectors, combined with well-designed and rigorous laboratory studies, could be crucial to determining their true role in disease transmission and guiding more effective and inclusive control strategies.

## 4. Conclusions

This review of the arboviruses found in Simuliidae reveals that, despite advances made in virus identification in other vectors, this group remains relatively unexplored compared to mosquitoes and ticks. There are few arboviruses that have been shown to have vector competence in Simuliidae. The use of metagenomics has proven to be a powerful tool for revealing viral diversity in hematophagous vectors, enabling the detection of both known and unknown viruses, including insect-specific viruses with potential implications for arbovirus evolution and transmission dynamics. As this technology continues to be applied to species within the Simuliidae family, our understanding of viromes in vectors and their role in arbovirus transmission will expand. This approach will not only enrich our knowledge of emerging and re-emerging viruses but will also contribute to understanding the ecology, evolution, and potential zoonotic threats of viruses harbored by Simuliidae. Public health will also be directly impacted through improved surveillance and the development of more effective strategies for controlling vector-borne pathogens that cause diseases in humans and animals.

## Figures and Tables

**Figure 1 life-15-00807-f001:**
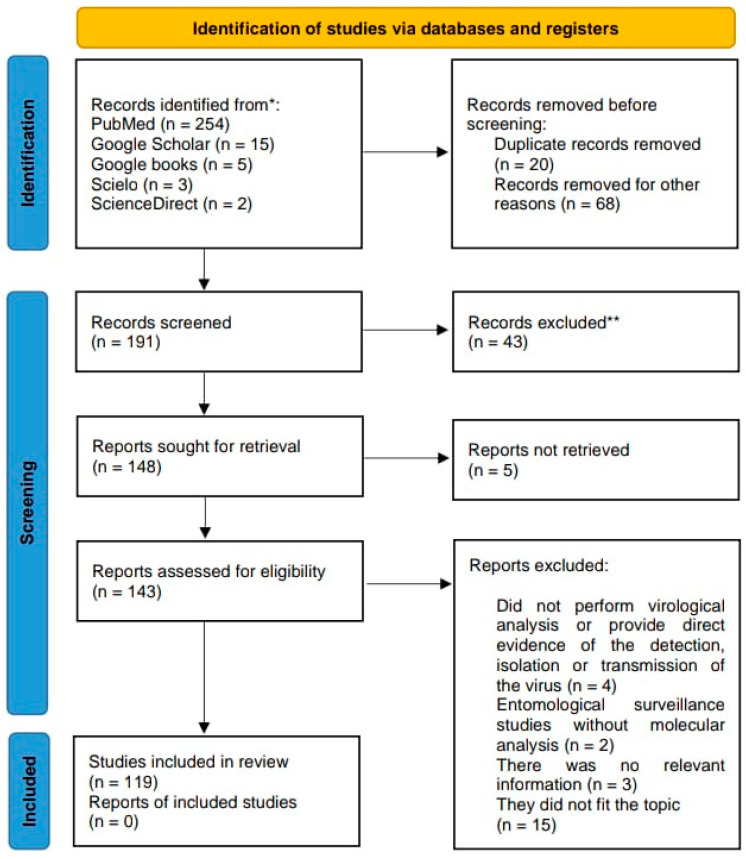
PRISMA analysis. Process of selecting scientific literature in a systematic review of arboviruses in Simuliidae using the PRISMA algorithm. The list of all records found with the mentioned keywords is given in the Appendix A. * Number of records identified from each database or register searched. ** Records excluded by a human; no automation tool was used.

**Figure 2 life-15-00807-f002:**
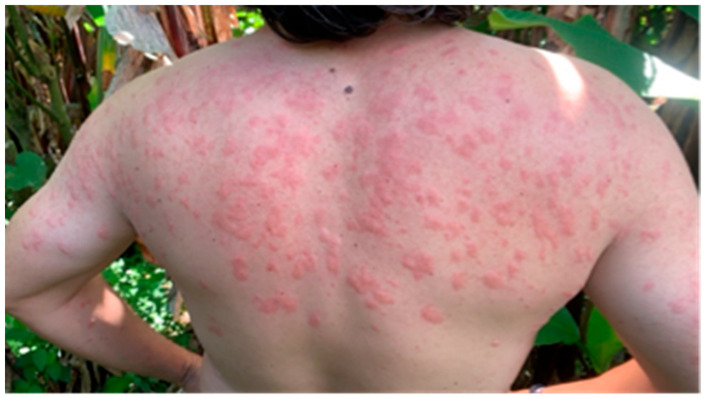
Severe black fly hypersensitivity in a person (pictured JRA-G), caused by multiple bites from *S. ochraceum* in Chiapas, Mexico, presenting a skin reaction with red spots and inflammation.

**Figure 3 life-15-00807-f003:**
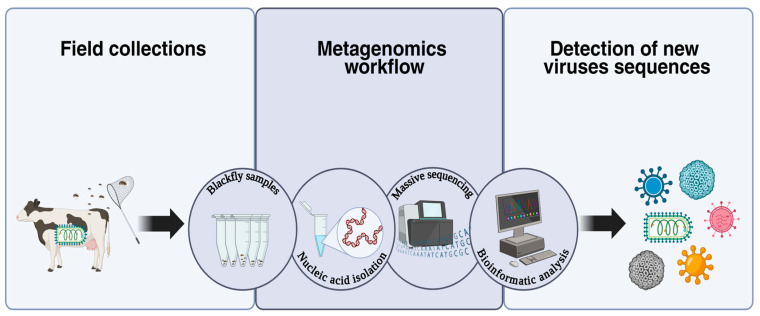
Schematic representation of the metagenomic workflow for pathogen detection in hematophagous arthropods. Biological samples are processed through nucleic acid extraction and next-generation sequencing (NGS). Subsequently, bioinformatics analysis enables the identification of novel pathogens, including viruses, bacteria, and other microorganisms of interest to public and veterinary health. Image created in Biorender (https://www.biorender.com accessed on 15 April 2025).

**Figure 4 life-15-00807-f004:**
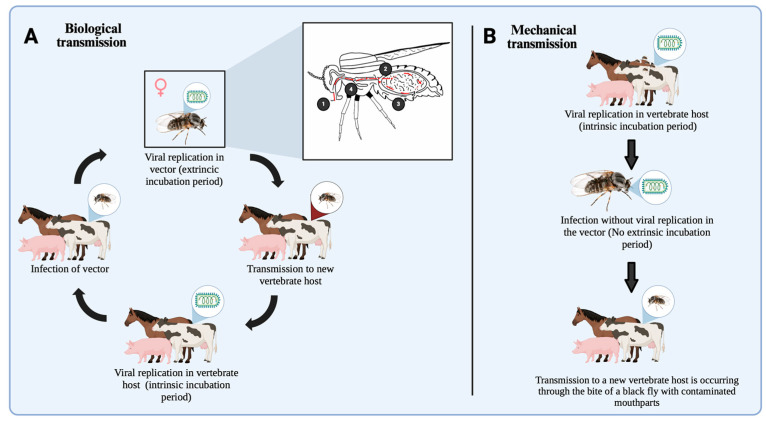
Mechanisms of horizontal transmission of arboviruses in Simuliidae. (**A**) In biological transmission, the ingested virus must cross via several physiological barriers within the black fly to establish a systemic infection. After ingestion, the virus must traverse the peritrophic matrix and penetrate the midgut epithelium (**1**), where it encounters the midgut infection barrier (MIB). If an infection is established in these cells, the virus must then overcome the midgut escape barrier (MEB) (**2**) to disseminate through the hemolymph and infect secondary tissues. Subsequently, it must reach the salivary gland infection barrier (SGIB) (**3**), and, finally, the salivary gland escape barrier (**4**), to reach the lumen and be secreted in the saliva. Once this process is completed, the virus is transmitted to a new vertebrate host during a subsequent blood meal, where viral replication occurs. (**B**) In mechanical transmission, the virus does not replicate within the vector. Transmission occurs passively via contaminated mouthparts, as the insect transfers viral particles directly from an infected host to another during feeding, without crossing physiological barriers or establishing infection in the black fly. Image created in Biorender (https://www.biorender.com accessed on 15 April 2025).

**Table 1 life-15-00807-t001:** Molecular characteristics of the seven viral families that include a representative sample of relevant arboviruses of medical and veterinary importance.

* Viral Family (Order)	Virion Size (nm)	Nuclei Acid Type	Number of Segments	Genome Size	Example of Arbovirus	References
*Flaviviridae* (*Amarillovirales*)	40–60 nm	ssRNA +	Not segmented	9–13 kb	Dengue virus, Zika virus, Yellow fever virus, Japanese encephalitis virus, Saint Louis encephalitis virus, Murray Valley encephalitis virus	[37,38,39,40]
*Togaviridae* (*Martellivirales*)	65–70 nm	ssRNA +	Not segmented	10–12 kb	Chikungunya virus, Ross River virus, O’nyong-nyong virus, Sindbis virus, Barmah Forest virus, Mayaro virus, Western, Eastern, and Venezuelan equine encephalitis viruses	[37,41,42,43]
*Peribunyaviridae* (*Elliovirales*)	80–120 nm	ssRNA −	Segmented (3 segments)	10.7–12.5 kb	La Crosse virus, Oropouche virus, Akabane virus	[30,44,45]
*Phenuiviridae* (*Hareavirales*)	80–120 nm	ssRNA −	Segmented (3 segments)	8.1–25.1 kb	Rift Valley fever virus	[46]
*Nairoviridae* (*Hareavirales*)	80–120 nm	ssRNA −	Segmented (3 segments)	17.2–21.1 kb	Crimean-Congo hemorrhagic fever virus	[47]
*Sedoreoviridae* (*Reovirales*)	60–85 nm	dsRNA	Segmented (10–12 segments)	18–30 kb	Bluetongue virus, African horse sickness virus, Equine encephalitis virus, Epizootic hemorrhagic disease virus	[48,49]
*Ortomixoviridae* (*Articulavirales*)	80–120 nm	ssRNA −	Segmented (6–8 segments)	10–15 kb	Thogoto virus Dhori virus	[50]
*Rhabdoviridae* (*Mononegavirales*)	100–180 nm	ssRNA −	Not segmented	10–16 kb	Vesicular stomatitis virus Bovine Ephemeral Fever	[51]
*Asfarviridae* (*Asfuvirales*)	175–215 nm	dsDNA	Not segmented	17–19 kb	African Swine Fever Virus	[52]

* The taxonomic nomenclature is based on the most recent classification according to the International Committee on the Taxonomy of Viruses.

**Table 2 life-15-00807-t002:** Methodologies used for the detection, isolation, or diagnosis of arboviruses in Simuliidae.

Virus Name	Family	Genome Size and Type	Methodology Used in the Report	Reference
Myxoma virus	*Poxviridae*	160 kb, dsDNA	In vivo studies	[53]
Vesicular stomatitis viruses (VSV) (Serotype New Jersey and Indiana)	*Rhabdoviridae*	~11 kb, −ssRNA	Cytopathic effect in Vero-M cells Titration with fluorescent antibodies Plaque assay	[57]
			Plaque assay Microplate neutralization Virus re-isolation in cell culture RT-PCR	[72]
			RT-PCR Sequencing	[55]
			Real-time RT-PCR Sequencing	[79]
Venezuelan equine encephalitis virus (VEEV)	*Togaviridae*	~11–12 kb, +ssRNA	Cell culture Complement fixation test (CFT)	[64]
Rift Valley fever virus (RVFV)	*Phenuiviridae*	~11.9 kb, −ssRNA	Virus isolation in animal models Histopathology of liver tissue Complement fixation test (CFT) Hemagglutination inhibition test (HI) Electron microscopy	[61]
Snowshoe hare virus (SSHV)	*Peribunyaviridae*	~12–13 kb, −ssRNA	Cell culture. Serological test	[68]
Unclassified bunyaviruses	Anteriormente *Bunyaviridae*	~12–13 kb, −ssRNA	Cell culture. Plaque reduction neutralization test (PRNT)	[69]

## Data Availability

Not applicable.

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
