# Peer review of "Viruses in Simuliidae: An Updated Systematic Review of Arboviral Diversity and Vector Potential"

_life, 2025, doi:10.3390/life15050807_

Round 1

Reviewer 1 Report

Comments and Suggestions for Authors

Rivera-Martinez et al provide an overview of published reports of viruses that have been detected in Simullidae. The report itself is welcome as it is useful to know the details of what is likely a significant under-representation of virus occurrence in black flies. However, there just isn’t much to go on. As the investigators communicate, ~90 of ~230 references were examined for details on viruses in black flies. Within those, the authors found that the overwhelming majority are simple descriptive studies, ie that PCR, experimental transmission studies, or RNASeq demonstrated the presence of one or more viruses in the fly sample. This is useful! However, the paper lacks a significant punch that might be developed more thoroughly. For example, the authors do not address other than very briefly the significance of not just showing the virus is present, but rather that the virus is BIOLOGICALLY associated with the fly (ie that it replicates in the fly) and that the fly is thus not just a casual but rather a causal partner in this relationship. In fact, I don’t think the authors ever address this vacuum in the field, which I strongly believe they should. Has there been a single scientific study that conclusively demonstrated this relationship? For example, Rift Valley was detected in 1953; that’s >70 years ago and no one has further investigated this (possible) relationship as casual vs causal? The authors should explicitly address this, and use the paper as a clarion call for greater emphasis from the field.

Beyond this, there are only minor issues as the paper has been quite well written:

Table 1: particularly as poxviruses, which are included in the text but not in a table, could be included in Table 1 or 2, it makes sense to me to condense/combine those two tables and ensure that all viruses are present.

Lines 244-7: more detail here is warranted – what does it mean that the viruses reach the salivary glands? How is this determined? What’s the significance (to a non-insect person)? For example, luteoviruses in aphids have been demonstrated to both causally and casually associate with salivary glands; for the former, they must cross the midgut barrier and move to the salivary glands, a pretty significant task relative to just uptake with a meal and reintroduction (with no amplification) in the latter.

The various associated paperwork sections (Instiutional Review Board, etc) are incorrect and should be addressed.

Author Response

Dear Reviewer: We would like to sincerely thank you for your thoughtful and constructive comments, which have greatly enriched our work. We truly appreciate the time and effort you dedicated to reviewing our manuscript.

Here are our answers: 

Comment 1:

Rivera-Martinez et al provide an overview of published reports of viruses that have been detected in Simullidae. The report itself is welcome as it is useful to know the details of what is likely a significant under-representation of virus occurrence in black flies. However, there just isn’t much to go on.

As the investigators communicate, ~90 of ~230 references were examined for details on viruses in black flies. Within those, the authors found that the overwhelming majority are simple descriptive studies, ie that PCR, experimental transmission studies, or RNASeq demonstrated the presence of one or more viruses in the fly sample.

This is useful! However, the paper lacks a significant punch that might be developed more thoroughly. For example, the authors do not address other than very briefly the significance of not just showing the virus is present, but rather that the virus is BIOLOGICALLY associated with the fly (ie that it replicates in the fly) and that the fly is thus not just a casual but rather a causal partner in this relationship.

In fact, I don’t think the authors ever address this vacuum in the field, which I strongly believe they should. Has there been a single scientific study that conclusively demonstrated this relationship?

For example, Rift Valley was detected in 1953; that’s >70 years ago and no one has further investigated this (possible) relationship as casual vs causal? The authors should explicitly address this and use the paper as a clarion call for greater emphasis from the field.

Beyond this, there are only minor issues as the paper has been quite well written.

  • Answer 1: We sincerely appreciate this observation. We have added a new section (3.6) that discusses the mechanisms of virus transmission in black flies from both causal and casual perspectives, taking as example VSVV and FRVV. We believe that your comments significantly enriched the manuscript.

Comment 2:

Table 1: particularly as poxviruses, which are included in the text but not in a table, could be included in Table 1 or 2, it makes sense to me to condense/combine those two tables and ensure that all viruses are present.

  • Answer 2: Table 2 has been updated to include the previously omitted viruses detected in Simuliidae. Tables 1 and 2 have not been merged or condensed, as they serve distinct purposes: Table 1 provides a general overview of arbovirus families, irrespective of their associated vectors, while Table 2 specifically focuses on viruses that have been detected or isolated in Simuliidae.

Comment 3:

Lines 244-7: more detail here is warranted – what does it mean that the viruses reach the salivary glands? How is this determined? What’s the significance (to a non-insect person)? For example, luteoviruses in aphids have been demonstrated to both causally and casually associate with salivary glands; for the former, they must cross the midgut barrier and move to the salivary glands, a pretty significant task relative to just uptake with a meal and reintroduction (with no amplification) in the latter.

  • Answer 3: A more detailed explanation has been provided concerning the vector competence experiments conducted by Austin (1967) to demonstrate the potential of black flies as vector to 10 arboviruses. Now the paragraph is clear and explain why some viruses tested were able to reach the salivary glands, highlighting their relevance in potential transmission as well. Also, regarding vector competence, this section has been further developed in response to Comment No. 1, where we discuss both the biological and mechanical transmission of arboviruses in Simuliidae.

Comment 4:

The various associated paperwork sections (Instiutional Review Board, etc) are incorrect and should be addressed.

  • Answer 4: Already corrected.

Finally, you will find attached the MS with track changes.

Reviewer 2 Report

Comments and Suggestions for Authors

"Viruses in Simuliidae: Where are we? -An Updated Review"

Overall the paper is not the most through review of the literature. There are gaps seemingly missing and I am not doing all of the research to fill them but show a fraction below.

General comments: Please work on the grammar. It is sloppy with family names in italics for example but genus names i.e. "Culex" not in italics.
Simuliidae should not be in italics it is a family name.
Additional grammar checks are needed by the editorial staff of the journal.

Lines 21-22 "have enabled the identification of non-cultivable viruses, greatly enhancing our understanding of black fly-borne viral diversity and their public and veterinary health implications."
True but it also lets scientists just detect viruses that are no transmittable in insects and label them as possible vectors incorrectly.

I would suggest that this review is missing a number of past real isolations of viruses rather than just molecular detection.
For example Adler Currie and Wood 2004 discuss the arbovirsues from North American black flies including isolations of an unnamed Bunyavirus, Eastern Equine Encephalitis, Snowshoe Hare virus, and some sort of unidentified arbovirus from Simulium meridionale. All discussed on page 108.
Adler, P.H.; Currie, D.C.; Wood, D.M. 2004. The Black Flies (Simuliidae) of North America; Cornell University
Press: Ithaca, NY, USA.

Alternatively Reeves and Milby 1990 looked at thousands of Simulium and did not isolate viruses from any.
Milby MM, Reeves WC. Natural infection in arthropod vectors. In: Reeves WC, editor. Epidemiology and control of mosquito-borne arboviruses in California, 1953–1987. Sacramento (CA): California Mosquito and Vector Control Association; 1990. p. 128–144.

Table 1: Why are the vector-borne Rhabdoviruses viruses such as Chandipura virus and Vesicular Stomatitis viruses , Ephemerovirus (Ephmermeral Fever), Sunrhavirus (Dillards Draw), etc missing? 
They are more significant than some on the list such as Dhori virus.

Figure 3: In the modern era where you can make digital images fairly easily why are you using mosquitoes in a paper about black flies?

Lines 182-183. There are numerous outbreaks with probable overwintering of VSV in the northern and central USA not just the southeastern USA. An example was the molecular detection in black flies, Simulium bivittatum, by Drolet et al 2021.
Drolet, B.S.; Reeves,W.K.; Bennett, K.E.; Pauszek, S.J.; Bertram, M.R.; Rodriguez, L.L. Identical Viral Genetic Sequence Found in Black Flies (Simulium bivittatum) and the Equine Index Case of the 2006 U.S. Vesicular Stomatitis Outbreak. Pathogens 2021, 10, 929. https://doi.org/10.3390/pathogens10080929

Lines 199-202: These statements are largely true about the USA also. NM, AZ, CO, WY, MT, and TX  all have black fly associated VSV outbreaks.

I think this paper is a decent review but it largely does not include much new information that was not reviewed in works such as Adler et al. 2004 and it is missing other references.

Comments on the Quality of English Language

See review comments.

Author Response

Dear Reviewer: 

We would like to sincerely thank you for your insightful and detailed comments. Your suggestions have significantly improved the quality of our work. All references you recommended were very helpful in completing and strengthening the information presented in the manuscript, now are more detailed. Thanks to your valuables comments, we believe the paper is now more complete and better structured.

Here are our answers to your comments: 

Comment 1

General comments: Please work on the grammar. It is sloppy with family names in italics for example but genus names i.e. "Culex" not in italics.

  • Answer 1: All genus names of insect are now in italics letters.

Comment 2

Simuliidae should not be in italics it is a family name.

  • Answer 2: All family names of insects are now in normal letters.

Comment 3

Additional grammar checks are needed by the editorial staff of the journal.

  • Answer 3: The grammar was again reviewed in detail.

Comment 4

Lines 21-22 "have enabled the identification of non-cultivable viruses, greatly enhancing our understanding of black fly-borne viral diversity and their public and veterinary health implications." True but it also lets scientists just detect viruses that are no transmittable in insects and label them as possible vectors incorrectly.

  • Answer 4: We improved the meaning of the sentence and explained that there are specific insect viruses that are not transmissible to vertebrates.

Comment 5

I would suggest that this review is missing a number of past real isolations of viruses rather than just molecular detection. For example Adler Currie and Wood 2004 discuss the arbovirsues from North American black flies including isolations of an unnamed Bunyavirus, Eastern Equine Encephalitis, Snowshoe Hare virus, and some sort of unidentified arbovirus from Simulium meridionale. All discussed on page 108.

Adler, P.H.; Currie, D.C.; Wood, D.M. 2004. The Black Flies (Simuliidae) of North America; Cornell University

Press: Ithaca, NY, USA.

Alternatively Reeves and Milby 1990 looked at thousands of Simulium and did not isolate viruses from any. 

Milby MM, Reeves WC. Natural infection in arthropod vectors. In: Reeves WC, editor. Epidemiology and control of mosquito-borne arboviruses in California, 1953–1987. Sacramento (CA): California Mosquito and Vector Control Association; 1990. p. 128–144.

Additional viruses isolated from female Simuliidae include Snowshoe hare virus in Simulium malyschevi, an uncharacterized Bunyavirus in Simulium bivittatum, and Eastern equine encephalitis virus in Simulium johannseni (Alder et al., 2004).

  • Answer 5: All the references you mentioned are already integrated into the text.

Comment 6

Table 1: Why are the vector-borne Rhabdoviruses viruses such as Chandipura virus and Vesicular Stomatitis viruses , Ephemerovirus (Ephmermeral Fever), Sunrhavirus (Dillards Draw), etc missing? They are more significant than some on the list such as Dhori virus.

  • Answer 6: The table you mentioned provides an overview of the molecular characteristics of arbovirus families and includes examples of some arboviruses. The revised version of the table now includes the viruses you kindly suggested.

Comment 7

Figure 3: In the modern era where you can make digital images fairly easily why are you using mosquitoes in a paper about black flies?

  • Answer 7: We have improved the figures. General changes were made to Figures 3 and 4 based on your comments as well as those from Reviewers 1 and 3. We think are now significantly better explained.

Comment 8

Lines 182-183. There are numerous outbreaks with probable overwintering of VSV in the northern and central USA not just the southeastern USA. An example was the molecular detection in black flies, Simulium bivittatum, by Drolet et al 2021.

Drolet, B.S.; Reeves,W.K.; Bennett, K.E.; Pauszek, S.J.; Bertram, M.R.; Rodriguez, L.L. Identical Viral Genetic Sequence Found in Black Flies (Simulium bivittatum) and the Equine Index Case of the 2006 U.S. Vesicular Stomatitis Outbreak. Pathogens 2021, 10, 929. https://doi.org/10.3390/pathogens10080929

Lines 199-202: These statements are largely true about the USA also. NM, AZ, CO, WY, MT, and TX all have black fly associated VSV outbreaks.

  • Answer 8: Thank you for pointing out the missing references. They have now been included in the text.

Comment 9

I think this paper is a decent review but it largely does not include much new information that was not reviewed in works such as Adler et al. 2004 and it is missing other references.

  • Answer: The information has been expanded and improved, and additional relevant citations that were previously overlooked have been included. Now, this work is more comprehensive.

Finally, you will find attached the MS with track changes.

Reviewer 3 Report

Comments and Suggestions for Authors

Author Response

Dear Reviewer:

We truly appreciate your valuable comments and thoughtful suggestions. In particular, the change you proposed to the title has greatly improved the focus and clarity of the manuscript.

Here are our answers for your comments: 

Comment 1

Title: Consider rephrasing for clarity and academic tone: Suggested Title: Viruses in Simuliidae: An Updated Systematic Review of Arboviral Diversity and Vector Potential

  • Answer 1: Already changed. We think that title proposed is better for this work.

Comment 2

Introduction:

Paragraph 1: Consider breaking this long paragraph into two for better readability.

  • Answer 2: Already done.

Comment 3

Sentence: "..such as onchocerciasis and mansonellosis, however.." - grammatically incorrect; should be a semicolon or separate sentences.

Suggested: "

...such as onchocerciasis and mansonellosis; however, their potential..."

  • Answer 3: Semicolon added, improved sentence.

Comment 4

Consider briefly explaining why previous studies overlooked Simuliidae e.g., ecological behavior, difficulty in colonization, or lack of outbreaks linked directly to them.

  • Answer 4: We added: “This is attributable to several factors, including the technical challenges associated with collecting these insects from their often restricted and remote habitats, the difficulty of establishing and maintaining stable colonies under laboratory conditions, and the limited feasibility of conducting vector competence studies (Adler et al. 2004; Cunze at al. 2024) [3-4]”.

Materials and Methods:

Comment 5

Excellent use of PRISMA protocol. Consider referencing the PRISMA 2020 checklist and indicating whether a supplementary file with the full checklist is provided. Provide specific date of the last search (e.g., "Search conducted up to January 20, 2025").

  • Answer 5: Thank you. The specific date and supplementary list of the full checklist of papers were included.

Comment 6

Conclusion:

Well-written and insightful. Sentence: "…..will also have a direct impact on public health.." could be strengthened by specifying how: e.g.,.. by improving early detection systems and informing vector control

strategies."

  • Answer 6: The sentence were changed by “Public health will also be directly impacted through improved surveillance and the development of more effective strategies for controlling vector-borne pathogens that cause diseases in humans and animals”.

Minor Language and Formatting Corrections:

Comment 7

Ensure consistency in using scientific names (e.g., Simulium italicized).

  • Answer 7: Checked and corrected.

Comment 8

Consider editing for passive voice to improve engagement in some parts.

  • Answer 8: Done.

Finally, you will find attached the MS with track changes.

Round 2

Reviewer 2 Report

Comments and Suggestions for Authors

There is very little new in this manuscript that has not already been published in textbooks.

I think the manuscript is improved with the edits.

I still find the Line 131 through 148 very odd.  Rhabdoviridae include a number of arboviruses and are probably the MOST COMMONLY ASSOCIATED ARBOVIRUSES WITH SIMULIIDAE. So why is there a significant discussion about assorted mosquito, Culicoides, and tick associated viruses.

In a past review I found several citations to arboviruses associates with Simuliidae. Not all are cited here and that makes me suspicious that even more are missing. However, I am not going to do the research for the authors.

I really do not see anything here that was not basically already in these two textbook chapters (oddly not cited)

Adler, P. H. 2005. Black flies, the Simuliidae. Pp. 127-140. In W. C. Marquardt (ed.). Biology of Disease Vectors, 2nd edition. Elsevier Academic Press, San Diego, CA.

Adler and McCreadie, 2019. Chapter 14 - black flies (Simuliidae) G.R. Mullen, L.A. Durden (Eds.), Medical and Veterinary Entomology (Third Edition), Academic Press (2019), pp. 237-259

Author Response

Dear Revisor,

Thank you very much for your comments, below you will find the answer to each of them: 

Comment 1: I still find the Line 131 through 148 very odd.  Rhabdoviridae include a number of arboviruses and are probably the MOST COMMONLY ASSOCIATED ARBOVIRUSES WITH SIMULIIDAE. So why is there a significant discussion about assorted mosquito, Culicoides, and tick associated viruses.

Answer 1:  Thank you for your observation. The section from lines 131 to 148 was intentionally written to provide a general overview of the importance of arboviruses, as reflected in the original title of that section, Relevance of Arbovirosis. The discussion begins, from a general point of view, highlighting that arboviruses transmitted by various arthropod vectors, such as mosquitoes, Culicoides, and ticks in order to provide the reader with a broader context regarding the diversity of viruses and their vectors. This general introduction was intended to set the stage for the more focused discussion on arboviruses associated with Simuliidae presented in the subsequent sections.

However, to improve clarity and avoid any potential confusion, we have shortened the section and changed the title for: General Relevance of Arbovirosis in Vector-Borne Diseases.

Comment 2: In a past review I found several citations to arboviruses associates with Simuliidae. Not all are cited here and that makes me suspicious that even more are missing. However, I am not going to do the research for the authors.

Answer 2: We carefully reviewed both the first and second versions of the manuscript. In this updated version that we have just uploaded, we added approximately 20 additional references that we had previously overlooked. Unfortunately, accessing classic entomological literature from Mexico is quite challenging. In the past, the website of the Armed Forces Pest Management Board was very helpful for this purpose; however, it is no longer available, which makes the literature search more difficult.

Comment 3: I really do not see anything here that was not basically already in these two textbook chapters (oddly not cited)

1) Adler, P. H. 2005. Black flies, the Simuliidae. Pp. 127-140. In W. C. Marquardt (ed.). Biology of Disease Vectors, 2nd edition. Elsevier Academic Press, San Diego, CA. 2) Adler and McCreadie, 2019. Chapter 14 - black flies (Simuliidae) G.R. Mullen, L.A. Durden (Eds.), Medical and Veterinary Entomology (Third Edition), Academic Press (2019), pp. 237-259.

Answer 3: Thank you for your comment. We appreciate your observation regarding the textbook chapters. Initially, we did not cite Adler, P.H. (2005) because, although the author mentions the presence of five arboviruses isolated from wild black flies, the specific viruses were not identified. While we fully acknowledge Dr. Peter Adler as a leading global authority on Simuliidae, we chose in the last version to exclude the reference due to the lack of detailed information. However, we have now included the citation in the uploaded version, as we consider it a classic and valuable reference in the literature.

Now, regarding Adler and McCreadie (2019), they cite the work of Palmer (1995), who noted that black flies were suspected of transmitting Rift Valley fever virus during an outbreak along the Orange River in 1975, we cited both, including the original paper of 1977.

Adler and McCreadie (2019) also reference the study by Chanteau et al. (1993), which implicated Simulium buissoni in the potential or hypothetical transmission of hepatitis B virus (HBV). We believe that the study by Chanteau et al. is important to include, as it offers valuable context and supports the discussion on potential zoonotic viruses found in black flies. Additionally, we have included recent studies on VSV. Although these refer to the same virus, we consider them relevant due to their distinct geographic origins and their contribution to illustrating the broad distribution of VSV.

Finally, we believe these revisions have significantly enhanced the overall quality and clarity of the manuscript. By incorporating additional references.

Attached is the new version of the manuscript. Thank you in advance for your valuable time and comments.
